# Joint Detection and Communication over Type-Sensitive Networks

**DOI:** 10.3390/e25091313

**Published:** 2023-09-08

**Authors:** Joni Shaska, Urbashi Mitra

**Affiliations:** Department of Electrical and Computer Engineering, University of Southern California, Los Angeles, CA 90089, USA; ubli@usc.edu

**Keywords:** heterogeneous networks, method of types, large-scale networks, information measures

## Abstract

Due to the difficulty of decentralized inference with conditional dependent observations, and motivated by large-scale heterogeneous networks, we formulate a framework for decentralized detection with coupled observations. Each agent has a state, and the empirical distribution of all agents’ states or the type of network dictates the individual agents’ behavior. In particular, agents’ observations depend on both the underlying hypothesis as well as the empirical distribution of the agents’ states. Hence, our framework captures a high degree of coupling, in that an individual agent’s behavior depends on both the underlying hypothesis and the behavior of all other agents in the network. Considering this framework, the method of types, and a series of equicontinuity arguments, we derive the error exponent for the case in which all agents are identical and show that this error exponent depends on only a single empirical distribution. The analysis is extended to the multi-class case, and numerical results with state-dependent agent signaling and state-dependent channels highlight the utility of the proposed framework for analysis of highly coupled environments.

## 1. Introduction

Decentralized detection is an important element in a wide range of modern applications, such as the Internet of Things [1], smart grids [2], cognitive radio [3], and millimeter-wave communications [4]. However, many classical results in decentralized detection assume that agents’ observations are independent, conditioned on the underlying hypothesis. This assumption fails to hold in many of these recent applications, such as human decision-making [5], sensor networks with correlated observations [6], and quorum sensing in microbial communities [7]. Unfortunately, the problem of decentralized detection with correlated observations is NP-Hard [8], and many of the classical results are not applicable in this case (for examples, see [9,10,11]). Recent work in decentralized detection has placed greater attention on the case of correlated observations [12,13,14,15]. Although recent advancements have been promising, the inherent difficulty of the problem has resulted in approximations and relaxations [13,15]. In this work, we build upon the state-dependent formulation introduced in [16] by allowing agents’ observations to depend on both the underlying hypothesis as well as the empirical distribution, or *type*, of their states. The notion of type has a rich history in information theory and statistics, being first introduced by Csiszar [17]. Today, the method of types has been further developed [18] and is used in a variety of fields, such as control [19], machine learning [20], statistics [21], and even DNA storage channels [22].

Conditionally correlated observations can be handled under specific signal models [15] and assumptions [12,16,23,24,25]. In particular, ref. [15] studied bandwidth-constrained detection under the Neyman–Pearson criterion and solved a relaxation of the problem. Several works [23,24,25] have studied the problem under communication constraints, with [23] showing that the network learns the hypothesis exponentially quickly under constrained [23] and randomized [24] communication. Moreocer, [25] developed a deep learning algorithm for real-time industry constraints. Other works have attempted to decouple agents’ observations via algorithms [13] and specific models [12,16,26]. In [12], a hidden variable was introduced that allows the observations of the agents to be independent, conditioned on the hidden variable, and it was proved that threshold-based decisions are optimal under certain model assumptions. Unfortunately, even if a problem of interest falls under this framework, the assumptions are rather strong and fail in a number of applications. In our prior work [16,26], we introduced a *state* variable for each agent and allowed the agents’ observations to be independent conditioned on both the hypothesis and the agent’s state. We proved similar results to those of [12,27] under much weaker conditions. However, the model proposed in [16,26] grants each agent its own individual state, whereas in [12] agents may share a common hidden variable.

The framework was extended in [16,26]; herein agents’ observations, *depend on a common variable*, i.e., the type of the agents’ states. In [16], it was assumed that agents know their individual state and that the fusion center knows the states of all agents. We strongly relax this assumption; agents do not know their state, and the fusion center only knows the empirical distribution of the agents’ states. Another key difference is that in [16,26] the state variable *is sufficient to decouple agents’ observations*, whereas in this work all agent states are necessary to decouple observations, allowing this formulation to handle stronger forms of coupling. The need for the empirical distribution calls for different analysis techniques from those in [12,16,26] via the method of types. We further introduce a communication link between the agents and the centralized decision-maker (called the *fusion center*) which is not present in [12,16,26].

Many works in decentralized detection include a communication link between the agents and the fusion center; the idea itself is not new [28,29,30]. However, in prior works the statistical properties of the communication channel were assumed to be independent of the network’s behavior. A contribution of our current work is that we *allow the quality of the communication channel to vary with the network’s behavior*. This is again accomplished by allowing the channel to vary with the type of the agents’ states. The concept of a channel with state-dependent noise has been previously considered in information theory [31], and is in use today [32,33,34]. However, most of the aforementioned works involving the notion of state have focused primarily on communication over channels with state, and have not examined joint detection and communication. While recent works on estimation exist, they were the context of estimating the channel state to improve communication performance [34,35,36]. Notably, signal-dependent noise [37] can be accommodated in our proposed model. In particular, these models are relevant to visible light communication [38], magnetic recording [39,40], and imaging applications [41,42].

As an example, we may consider the occurrence of such forms of coupling in microbial systems. Microbial communities synthesize signaling molecules [7]; when sensed in the environment, these can result in individual gene expressions that lead to new collective behaviors through a process called quorum sensing. Specifically, cell *i* only engages in quorum sensing when the received number of autoinducer molecules from the environment $A_{i}$ exceeds a certain threshold $\tau_{A}$. A common model involves assuming that $A_{i}$ follows a Poisson distribution conditioned on the total number of *synthases* (synthases are enzymes within a cell that are responsible for the production of autoinducer molecules) in the community and the number of receptors in a cell *i*, provided as $R_{i}$ [43]:$$P\left(A_{i}=k|R_{i},S_{1},\ldots ;S_{n}\right)=\frac{{\left(\lambda R_{i}\sum_{j=1}^{n}S_{j}\right)}^{k}\exp\left(-\lambda R_{i}\sum_{j=1}^{n}S_{j}\right)}{k!}$$
where $S_{i}$ is the number of synthases present in cell *i* and λ>0 is a normalizing term. Hence, we can think of the number of synthases and receptors in cell *i* as being the **state** of cell *i*. Then, the observation of cell *i* depends on *the states of all other cells* through the summed total of synthases across the cells. This example illustrates the need for the current approach, as the models proposed in [12,16] cannot handle this form of coupling and do not lead to tractable asymptotic results.

In this work, we derive the *error exponent* as the network size grows. Assuming that all priors are known, the optimal asymptotic decay rate of the probability of error is provided by the *Chernoff information* [27,44,45], regardless of whether conditional independence holds. Using the Chernoff information, ref. [27] proved that identical rules are *asymptotically optimal* for identical agents, while [45] showed that identical binary quantizers are asymptotically optimal in power-constrained networks. The works in [27,45] both relied on conditional independence. A contribution of the present study is to remove the need for conditional independence through the development of a measure that is asymptotically equivalent to the Chernoff information and tractable in our scenario. The primary argument comes from the method of types, which, combined with a series of equicontinuity arguments, shows that asymptotic performance is dominated *by a single distribution*. Surprisingly, this dominating distribution is generally *not the true distribution of the agents’ states*.

Using the network type to decouple agents’ observations can be extended beyond pure decentralized detection. For instance, consensus algorithms used in blockchain applications often need to deal with faulty or nonconforming nodes [46]. Hence, it is possible to consider whether the node is conforming or not as the state and the total percentage of conforming nodes as the network type. Then, the problems of jointly estimating the network type (the consensus problem) and detecting the underlying hypothesis (the detection problem) can be considered. Much of the structure herein applies to such problems, as observations received by agents depend on the other agents’ states. Moreover, the hypothesis and network type are correlated; when more agents are faulty or nonconforming, an attack is more likely to be present.

Our contributions in this paper are as follows:We formulate a framework for distributed inference in which the agents’ observations are correlated through both the hypothesis and the empirical distribution (or *type*) of the network state. This formulation captures a high level of coupling between agents.We consider a distributed inference problem with a communication link between the agents and the fusion center, with the additional caveat that the noise over the link is dependent on the agents. Hence, our framework captures joint sensing with correlated observations as well as joint communications with correlated noise.We derive expressions for the error exponent for a single class of agents, then extend our results to the case of heterogeneous groups of identical agents. In particular, assuming that identical agents use a common rule, the optimal error exponent depends only on the ratios of the groups, not on the actual size of the groups themselves. This allows a wide range of problems to be studied in which there are multiple classes of agents that interfere with each other.We present a numerical example for a three-class case to highlight the utility of the proposed expression for the error exponent. In particular, we show how this expression can be used to optimize the ratios of heterogeneous groups in the presence of cross-class interference. This example further illustrates the fact that the true distribution may not dominate the asymptotics. The effect of the channel is observed as well.

### Notation

Random variables are denoted by capital letters *X* and specific realizations are denoted as lowercase letters *x*. Random vectors are denoted as boldface capital letters X and specific realizations are denoted with lowercase boldface letters x. Given a random vector (realization) X (x), $X^{\setminus k}$ ($x^{\setminus k}$) denotes the vector X (x) with the *k*th element removed. Calligraphic letters X denote sets. The symbol P denotes probabilities of events, and $E_{X}$ denotes expectations with respect to the random variable *X*.

## 2. Materials and Methods

The details concerning how plots are generated are provided in Section 5, along with a discussion of a specific example.

## 3. Problem Formulation, Definitions, and Assumptions

### 3.1. Problem Setup

Consider a set of *n* agents. The global environmental variable *H* is the binary H∈{0,1}. Agent *k* (k=1,2,…;n) receives a *signal* $Y_{k}\in Y$, with Y being the signal space. The probability density of $Y_{k}$ conditioned on H=h is denoted as $p_{h}^{k}\left(y\right)$. In addition, each agent takes a *state* $S_{k}\in \left\{1,2,\ldots ;m\right\}$, where *m* is a finite integer. The prior for the state of agent *k* is $p^{k}\left(s\right)$, and we define the vector $p^{k}={\left[p^{k}\left(1\right),\ldots ,p^{k}\left(m\right)\right]}^{T}$. The collection of states $S={\left[S_{1},\cdots ,S_{n}\right]}^{T}$ is called the *network state*, with joint density p(s). For a given network state, we denote the empirical distribution (or the *type*) of S as $Q_{S}^{n}$, that is,
(1)$$Q_{S}^{n}\left(i\right)=\frac{1}{n}\sum_{k=1}^{n}1_{\left\{S_{k}=i\right\}},\;i\in \left\{1,\ldots ;m\right\},$$
where $1_{\left\{S_{k}=i\right\}}$ is the indicator that agent *k* is in state *i*. Let $Q^{n}$ denote the set of all empirical distributions corresponding to sequences of length *n*; then, for a given $q^{n}\in Q^{n}$, $T(q^{n})$ is the *type-class of* $q^{n}$, i.e.,
(2)$$T\left(q^{n}\right)=\left\{s\in {\left\{1,2,\ldots ;m\right\}}^{n}:Q_{s}^{n}=q^{n}\right\},$$
where ${\left\{1,2,\ldots ;m\right\}}^{n}$ is the Cartesian product of {1,2,…;m} with itself *n* times. Note that $Q_{S}^{n}$ is a random vector with realization $q^{n}$, that is,
(3)$$p\left(q^{n}\right)=P\left(Q_{S}^{n}=q^{n}\right)=P\left(T\left(q^{n}\right)\right)=\sum_{s\in T(q^{n})}p\left(s\right).$$ The joint probability distribution of $Y_{k}$ and the network type under hypothesis H=h is provided by $p_{h}^{k}\left(y,q^{n}\right)$. The associated conditional density is denoted as $p_{h}^{k}\left(y|q^{n}\right)$. Let $P^{m}$ denote the probability simplex in $R^{m}$:(4)$$P^{m}=\left\{q\in R^{m}:q_{i}\geq 0,\sum_{i}q_{i}=1\right\}.$$ For $q\in P^{m}$, the conditional density $p_{h}^{k}\left(y|q\right)$ is called the *signal model for agent* k. When we write densities conditioned on $q\in P^{m}$, we are assuming that these densities have a functional dependence on q in order to avoid issues with measurability, as certain types may not be observable regardless of the size of the network. For a simple example, consider ${\left[\frac{1}{e},1-\frac{1}{e}\right]}^{T}\in P^{2}$, which is never in $Q^{n}$ for any *n* due to the fact that $\frac{1}{e}$ is irrational. We define $Y={\left[Y_{1},\cdots ,Y_{n}\right]}^{T}$, while the joint density of Y and $Q^{n}$ and the density of Y conditioned on $Q^{n}=q^{n}$ under H=h are denoted as $p_{h}\left(y,q^{n}\right)$ and $p_{h}\left(y|q^{n}\right)$, respectively. The joint density $p_{h}\left(y|q\right)$ is called the *joint signal model*. For brevity, we call the conditional distribution $p(H=h|q^{n})$ the *hypothesis model*. It is important to note that we *do not assume conditional independence of the agents’ observations*, i.e., we can have $p_{h}\left(y\right)\neq \prod_{k}p_{h}^{k}\left(y_{k}\right)$ for h∈{0,1}; we do, however, assume that the structure described below holds.

**Assumption** **1.**
*The joint signal model obeys the following: ∀y, $\forall q\in P^{m}$, ∀h∈{0,1},*

(5)
$$p_{h}\left(y|q\right)=\prod_{k=1}^{n}p_{h}^{k}\left(y_{k}|q\right).$$



Equation (Equation 5) states that the signal $Y_{k}$ of agent *k* is independent of $Y^{\setminus k}$ conditioned on *both H* and $Q^{n}$.

**Assumption** **2.**
*∀y, $\forall q^{n}\in Q^{n}$, h∈{0,1}, $p_{h}\left(y,q^{n}\right)>0$; the joint densities have the same support under both hypotheses.*


Upon receiving observation $Y_{k}$, agent *k* makes a *decision* $U_{k}\in \left\{1,2,\ldots ;b\right\}$ according to a *rule*, which is a (possibly randomized) function from Y to the decision space U. We denote the possibly randomized rule used by agent *k* as $\gamma_{k}$,
(6)$$U_{k}=\gamma_{k}\left(Y_{k}\right)∼p^{k}\left(u|y\right).$$ The collection of rules $\gamma ={\left[\gamma_{1},\cdots ,\gamma_{n}\right]}^{T}$ is called a *strategy*. After agent *k* has made its decision, it sends $U_{k}$ to the *fusion center* through a noisy communication link which is allowed to depend on the type $q^{n}$. Upon sending $U_{k}$, the fusion center receives the message $X_{k}$ with
(7)$$X_{k}∼p^{k}\left(x|u,q^{n}\right).$$ Given $q\in P^{m}$, the conditional density $p^{k}\left(x|u,q\right)$ is the *channel model for agent* k. We define $X={\left[X_{1},\cdots ,X_{n}\right]}^{T}$, and the joint conditional density p(x|u,q) is called the *joint channel model*.

**Assumption** **3.**
*The joint channel model obeys the following: ∀x, u, $\forall q\in P^{m}$,*

(8)
$$p\left(x|u,q\right)=\prod_{k=1}^{n}p^{k}\left(x_{k}|u_{k},q\right).$$



**Assumption** **4.**
*∀x, u, $\forall q\in P^{m}$, p(x|u,q)>0.*


The fusion center does not know the network state S, however, *we assume that it knows* $Q_{S}^{n}$. This assumption is not strong, as the empirical distribution $Q_{S}^{n}$ can be estimated via consensus methods [47]. Upon receiving messages X and $Q_{S}^{n}$, the fusion center makes an inference as to which hypothesis is true, denoted by $\hat{H}$. We seek to minimize the asymptotic decay rate of the probability of the error (as defined in Equation (Equation 10)). We assume that the fusion center is using the maximum a posterori (MAP) rule, i.e.,
(9)$$\hat{H}=\left\{\begin{array}{cc} 1, & \left(x,q^{n}\right)\in A^{\gamma }\\ 0, & \left(x,q^{n}\right)\in A^{\gamma^{c}} \end{array}\right.,\;\mathrm{where}\;A^{\gamma }=\left\{\left(x,q^{n}\right):\frac{p_{1}\left(x|q^{n}\right)}{p_{0}\left(x|q^{n}\right)}\geq \frac{p(H=0|q^{n})}{p(H=1|q^{n})}\right\},$$
which minimizes the probability of error for a given strategy γ. The set $A^{\gamma }$ depends on the specific strategy γ selected; given γ, it is possible to compute the optimal inference rule as a deterministic function of γ using Equation (Equation 9). The complete problem setup is summarized in Figure 1.

### 3.2. Definitions

We now introduce several definitions and concepts that are used throughout the paper.

**Definition** **1.***Let $P^{\gamma }\left(\hat{H}\neq H\right)$ be the probability of error under strategy γ. We define the error exponent* Λ *(provided the limit exists) as:*
(10)$$\Lambda =-\lim_{n\to \infty }\underset{\gamma }{\inf}\frac{1}{n}\log P^{\gamma ,\psi }\left(\hat{H}\neq H\right).$$

The limit Λ depends on the strategy γ. Thus, the strategy $\gamma^{*}$ that achieves the infimum may be such that the limit does not exist. Moreover, (Equation 10) makes no assumption as to how the statistical properties of the agents vary with *n*; in general, it is not possible to say anything about the existence of Λ. However, in many practical settings, such as homogeneous networks and power-constrained networks, Λ exists and has a nice closed-form solution [16,27,45]. The main result of this work is an equivalent characterization of the error exponent defined above, showing that in our scenario the limit does exist. This equivalent expression has several desirable properties, and we can directly optimize the equivalent expression.

**Definition** **2.**
*The Kullback–Leibler Divergence between two distributions q and p is provided as follows:*

$$D\left(q||p\right)=\sum_{x}q\left(x\right)\log\frac{q(x)}{p(x)}.$$



Here, we are interested in understanding the interactions between different *classes* of agents, where members of a given class are *identical*, defined as follows.

**Definition** **3.**
*Given a collection of n agents, these agents are identical if the following conditions hold:*
*1*.
*$p_{h}^{k}\left(y|q\right)=p_{h}^{j}\left(y|q\right)$ for all k,j∈{1,2,…;n}, h∈{0,1}, y∈Y, $q\in P^{m}$.*
*2*.
*$p^{k}\left(x|u,q\right)=p^{j}\left(x|u,q\right)$ for all k,j∈{1,2,…;n}, x∈X, u∈{1,2,…;b}, $q\in P^{m}$.*
*3*.
*The agent states $S_{k}$ are i.i.d. a priori, i.e., $p\left(s\right)=\prod_{k}p^{k}\left(s_{k}\right)=\prod_{k}p\left(s_{k}\right)$.*



Condition (1) states that, conditioned on the hypothesis *H* and the network type $Q_{S}^{n}$, the probability distributions on the received signals for all agents are the same. Similarly, Condition (2) states that, conditioned on the network type $Q_{S}^{n}$ and $U_{k}=u$ for all k∈{1,2,…;n}, the probability distributions on the received messages are the same for all agents.

**Definition** **4.**
*A class is a collection of agents that are all identical.*


### 3.3. Key Assumptions

We first derive the error exponent for the single-class case in Theorem 1, which is then generalized to the case of multiple classes.

**Assumption** **5.**
*Our key assumptions for Theorem 1 are as follows:*
*(a)* 
*All agents are identical, as provided in Definition 3. Hence, we remove the notational dependence on k in the sequel.*
*(b)* 
*The hypothesis model obeys the following:*

(11)
$$\lim_{n\to \infty }\frac{1}{n}\log\min_{q^{n}}\min\{p\left(H=1|q^{n}\right),p\left(\left(H=0|q^{n}\right)\right\}=0.$$

*(c)* 
*The signal model is continuous in q for all agents, that is, if ${\left\{\alpha_{i}\right\}}_{i\in Z}$ is a sequence in $P^{m}$ such that $\lim_{i\to \infty }\alpha_{i}=q$, then ∀y,*

(12)
$$\lim_{i\to \infty }p_{h}\left(y|\alpha_{i}\right)=p_{h}\left(y|q\right),\;h\in \left\{0,1\right\}.$$

*(d)* 
*The channel model is continuous in q for all agents. That is, if ${\left\{\alpha_{i}\right\}}_{i\in Z}$ is a sequence in $P^{m}$ such that $\lim_{i\to \infty }\alpha_{i}=q$, then ∀x, ∀u,*

(13)
$$\lim_{i\to \infty }p\left(x|u,\alpha_{i}\right)=p\left(x|u,q\right).$$




**Remark** **1.**
*Recall that the fusion center knows the empirical distribution $q^{n}$ and that the optimal rule is provided by (Equation 9). Hence, if (Equation 33) does not hold, then the threshold $\frac{p(H=0|q^{n})}{p(H=1|q^{n})}$ may either grow or decay exponentially quickly, biasing the fusion center to the point that the decisions u become irrelevant. Hence, if the empirical distribution of the state carries too much information about the hypothesis, then the probability of error can be driven to zero exponentially quickly by simply looking at the network state, regardless of the rules used by the agents, leading to the need for Assumption 5.b. Assumptions 5.c and 5.d imply that if two distributions in $P^{m}$ are close with respect to the standard Euclidean metric, then the resulting signal and channel models should be close for all y and x, respectively.*


## 4. Main Results and Important Corollaries

We first consider the single-class result (Theorem 1). We discuss its implications and outline the needed proof techniques, then turn our attention to the multi-class case, which begins by extending Theorem 1 to Lemma 1 and then stating Theorem 2 and its implications. For the main theorems, we provide proof outlines in this section and the complete proofs in Section 6. The extension of Theorem 1 to Lemma 1 is provided in Section A.2.

### 4.1. Single-Class Results

**Theorem** **1.**
*Subject to Assumptions 5.a–5.d,*

(14)
$$\Lambda =-\lim_{n\to \infty }\underset{\gamma }{\inf}\min_{\lambda \in [0,1]}\max_{q\in P^{m}}-D\left(q||p\right)+\frac{1}{n}\log\int_{x}p_{0}{\left(x|q\right)}^{1-\lambda }p_{1}{\left(x|q\right)}^{\lambda },$$

*where D(q||p) is the Kullback–Leibler (KL) divergence between the distribution $q\in P^{m}$ and the true state distribution p.*


Theorem 1 provides an alternative asymptotically equivalent expression for the error exponent. In particular, Theorem 1 states that a single distribution dominates the asymptotic performance. Interestingly, the dominating distribution is in general *not the true distribution of the agents’ states*, despite the fact that the empirical distribution of the states converges towards the true distribution. We then extend Theorem 1 to multiple classes; if agents with a single class use a common rule, the error exponent for each class depends only on the ratios of numbers of agents between classes.

We underscore why the Chernoff information is challenging to compute for our problem framework:(15)$$\Lambda =-\lim_{n\to \infty }\underset{\gamma }{\inf}\min_{\lambda \in [0,1]}\frac{1}{n}\log\int_{x}\sum_{q^{n}}p_{0}{\left(x,q^{n}\right)}^{1-\lambda }p_{1}{\left(x,q^{n}\right)}^{\lambda }.$$ As *n* grows, so does the space of potential strategies γ, possible messages x, and possible types $Q^{n}$. Even if we have identical agents using the same rule, the complexity and coupling due to the summation over $q^{n}$ remains. If agents use the same rule
(16)$$\begin{array}{cc} \frac{1}{n}\log\int_{x}\sum_{q^{n}}p_{0}{\left(x,q^{n}\right)}^{1-\lambda } & p_{1}{\left(x,q^{n}\right)}^{\lambda }=\frac{1}{n}\log\int_{x}\sum_{q^{n}}p_{0}{\left(x|q^{n}\right)}^{1-\lambda }p_{1}{\left(x|q^{n}\right)}^{\lambda }p{\left(q^{n}|H=0\right)}^{1-\lambda }p{\left(q^{n}|H=1\right)}^{\lambda } \end{array}$$
(17)$$\begin{array}{cc} \; & \;\;\;\;\;\;\;=\frac{1}{n}\log\sum_{q^{n}}p{\left(q^{n}|H=0\right)}^{1-\lambda }p{\left(q^{n}|H=1\right)}^{\lambda }\int_{x}\prod_{k}p_{0}^{k}{\left(x_{k}|q^{n}\right)}^{1-\lambda }p_{1}^{k}{\left(x_{k}|q^{n}\right)}^{\lambda } \end{array}$$
(18)$$\begin{array}{cc} \; & \;\;\;\;\;\;\;=\frac{1}{n}\log\sum_{q^{n}}p{\left(q^{n}|H=0\right)}^{1-\lambda }p{\left(q^{n}|H=1\right)}^{\lambda }\prod_{k}\int_{x}p_{0}^{k}{\left(x|q^{n}\right)}^{1-\lambda }p_{1}^{k}{\left(x|q^{n}\right)}^{\lambda } \end{array}$$
(19)$$\begin{array}{cc} \; & \;\;\;\;\;\;\;\overset{\left(a\right)}{=}\frac{1}{n}\log\sum_{q^{n}}p{\left(q^{n}|H=0\right)}^{1-\lambda }p{\left(q^{n}|H=1\right)}^{\lambda }{\left(\int_{x}p_{0}^{1}{\left(x|q^{n}\right)}^{1-\lambda }p_{1}^{1}{\left(x|q^{n}\right)}^{\lambda }\right)}^{n}, \end{array}$$
where (a) is due to the fact that agents are identical and use the same rule, then all terms in the product are identical. Note that due to the summation over $q^{n}$, as previously stated, the complexity of calculating the Chernoff information grows with *n*, leading to the need for Theorem 1.

There are a few key remarks that must be made here about Theorem 1:The maximization occurs over $P^{m}$ instead of $Q^{n}$; hence, we have directly removed the dependence on $q^{n}$. Because the expression in Theorem 1 is continuous over the compact set $P^{m}$, it always achieves its maximum (versus supremum). This is due to Assumptions 5.c and 5.d.Note that the second term is the classical Chernoff information corresponding to the fixed distributions $p_{h}\left(x|q\right)$, h∈{0,1} and that the KL divergence term can be thought of as a bias. Hence, we only need to consider the *m*-dimensional probability vector that yields the worst Chernoff information biased by the KL divergence. In a certain sense, q is sufficiently close to the true state distribution *p*, such that its poor performance (under strategy γ) cannot be ignored even in asymptotically large networks. *Only one distribution in* $P^{m}$ *dominates the asymptotic performance*, as expected, although it may not be the true distribution *p*. An instantiation of this is provided in the numerical results.The maximization for q takes place over all of $P^{m}$; however, it is only necessary to search a subset of $P^{m}$ to find the maximum, thereby reducing the computational cost. To determine the subset of interest, observe that
(20)$$\begin{array}{cc} \min_{q\in P^{m}}D\left(q||p\right)-\frac{1}{n}\log\int_{x}p_{0}{\left(x|q\right)}^{1-\lambda }p_{1}{\left(x|q\right)}^{\lambda } & \leq D\left(p||p\right)-\frac{1}{n}\log\int_{x}p_{0}{\left(x|p\right)}^{1-\lambda }p_{1}{\left(x|p\right)}^{\lambda } \end{array}$$
(21)$$\begin{array}{cc} & \;\;\;\;\;\;\;\;\;\;\;\;\;=-\frac{1}{n}\log\int_{x}p_{0}{\left(x|p\right)}^{1-\lambda }p_{1}{\left(x|p\right)}^{\lambda } \end{array}$$
(22)$$\begin{array}{cc} & \;\;\;\;\;\;\;\;\;\;\;\;\overset{\left(a\right)}{\leq }-\frac{1}{n}\log\sum_{u}p_{0}{\left(u|p\right)}^{1-\lambda }p_{1}{\left(u|p\right)}^{\lambda } \end{array}$$
(23)$$\begin{array}{cc} & \;\;\;\;\;\;\;\;\;\;\;\;\overset{\left(b\right)}{\leq }-\frac{1}{n}\log\int_{y}p_{0}{\left(y|p\right)}^{1-\lambda }p_{1}{\left(y|p\right)}^{\lambda }, \end{array}$$
where both (a) and (b) are due to Hölder’s inequality. Using the fact that the Chernoff information is non-negative [44], it can be seen that the distribution $q^{*}$ that achieves the maximum over $P^{m}$ must satisfy
(24)$$D(q^{*}\left||p\right)\leq -\frac{1}{n}\log\int_{y}p_{0}{\left(y|p\right)}^{1-\lambda }p_{1}{\left(y|p\right)}^{\lambda }=c\left(\lambda ,p\right).$$The right-hand side of (Equation 24) is the Chernoff information for the signal model under distribution *p*; hence, the maximizing $q^{*}$ must live in a ball defined by the Kullback–Leibler divergence centered at the distribution *p* with radius c(λ,p), thereby reducing the search space for the optimization. In fact, the Chernoff information admits a closed-form solution for a wide range of distributions, such as members of the exponential family [48]The expression in Theorem 1 admits the following property: for all $q\in P^{m}$ and λ∈[0,1],
(25)$$\begin{array}{cc} \frac{1}{n}\log\int_{x}p_{0}{\left(x|q\right)}^{1-\lambda }p_{1}{\left(x|q\right)}^{\lambda } & \overset{\left(a\right)}{=}\frac{1}{n}\log\int_{x}{\left[\prod_{k=1}^{n}p_{0}^{k}\left(x_{k}|q\right)\right]}^{1-\lambda }{\left[\prod_{k=1}^{n}p_{1}^{k}\left(x_{k}|q\right)\right]}^{\lambda } \end{array}$$
(26)$$\begin{array}{cc} =\frac{1}{n}\log\prod_{k=1}^{n}\int_{x_{k}}p_{0}^{k}{\left(x_{k}|q\right)}^{1-\lambda }p_{1}^{k}{\left(x_{k}|q\right)}^{\lambda }\; & =\frac{1}{n}\sum_{k=1}^{n}\log\int_{x_{k}}p_{0}^{k}{\left(x_{k}|q\right)}^{1-\lambda }p_{1}^{k}{\left(x_{k}|q\right)}^{\lambda },\;\;\;\; \end{array}$$
where (a) holds due to both Equations (Equation 5) and (Equation 8). Then, for agents using a common rule, all terms in the sum are equal; thus,
(27)$$\frac{1}{n}\log\int_{x}p_{0}{\left(x|q\right)}^{1-\lambda }p_{1}{\left(x|q\right)}^{\lambda }=\log\int_{x}p_{0}^{1}{\left(x|q\right)}^{1-\lambda }p_{1}^{1}{\left(x|q\right)}^{\lambda },$$
which does not depend on *n*, helping to simplify analysis.

We next sketch the proof of Theorem 1. We start from the classical Chernoff information and use it to show that
(28)$$\Lambda =-\lim_{n\to \infty }\underset{\gamma }{\inf}\min_{\lambda \in [0,1]}\frac{1}{n}\log\int_{x}\sum_{q^{n}}p_{0}{\left(x|q^{n}\right)}^{1-\lambda }p_{1}{\left(x|q^{n}\right)}^{\lambda }p\left(q^{n}\right).$$ To prove the result, we wish to show that
(29)$$|\frac{1}{n}\log\frac{\int_{x}\sum_{q^{n}}p_{0}{\left(x|q^{n}\right)}^{1-\lambda }p_{1}{\left(x|q^{n}\right)}^{\lambda }p\left(q^{n}\right)}{\max_{q\in P^{m}}\int_{x}p_{0}{\left(x|q\right)}^{1-\lambda }p_{1}{\left(x|q\right)}^{\lambda }2^{-nD(q||p)}}|<\epsilon .$$
*uniformly in λ and γ*, that is, we wish to show that for any ϵ>0 there exists an integer $n_{\epsilon }$ *that is independent of λ and γ* such that (Equation 29) holds for all $n\geq n_{\epsilon }$. Uniform convergence in λ and γ enables determination of the minimum and infimum, respectively, yielding
(30)$$\begin{array}{cc} \underset{\gamma }{\inf}\min_{\lambda \in [0,1]} & \frac{1}{n}\log\int_{x}\sum_{q^{n}}p_{0}{\left(x|q^{n}\right)}^{1-\lambda }p_{1}{\left(x|q^{n}\right)}^{\lambda }p\left(q^{n}\right)\\ \; & -\underset{\gamma }{\inf}\min_{\lambda \in [0,1]}\max_{q\in P^{m}}-D\left(q||p\right)+\frac{1}{n}\log\int_{x}p_{0}{\left(x|q\right)}^{1-\lambda }p_{1}{\left(x|q\right)}^{\lambda }\to 0, \end{array}$$
as n→∞, which is the desired assertion. Equivalently, it can be shown that
(31)$$\left(1-\epsilon \right)<{\left[\frac{\int_{x}\sum_{q^{n}}p_{0}{\left(x|q^{n}\right)}^{1-\lambda }p_{1}{\left(x|q^{n}\right)}^{\lambda }p\left(q^{n}\right)}{\max_{q\in P^{m}}\int_{x}p_{0}{\left(x|q\right)}^{1-\lambda }p_{1}{\left(x|q\right)}^{\lambda }2^{-nD(q||p)}}\right]}^{\frac{1}{n}}<\left(1+\epsilon \right),$$
uniformly in λ and γ.

### 4.2. Multi-Class Results

We now discuss extending the results in the previous section to the case of multiple classes. Consider a set of $n_{c}<\infty$ classes. For a given class $c\in \{1,2,\ldots ;n_{c}\}$, let $c_{k}$ be the number of agents that belong to class *c*. Then, let $Y_{c,k}\in Y$ and $S_{c,k}\in \left\{1,2,\ldots ;m\right\}$ be the signal and state, respectively, of the *k*th agent in class *c*. Without loss of generality, assume that the signal space Y and state space {1,2,…;m} are the same for all classes. Furthermore, for a given network state S, let $Q_{c,S}^{n}$ denote the type of the states of the agents belonging to class *c*, i.e.,
(32)$$Q_{c,S}^{n}\left(i\right)=\frac{1}{c_{k}}\sum_{k=1}^{c_{k}}1_{\left\{S_{c,k}=i\right\}},\;i\in \left\{1,\ldots ;m\right\}.$$ For given realizations of the class types $q_{1}^{n},\ldots ;q_{n_{c}}^{n}$, the signal model and state prior for class *c* are denoted as $p^{c}\left(y|q_{1}^{n},\ldots ;q_{n_{c}}^{n}\right)$ and $p^{c}\left(s\right)$, respectively, and with $p^{c}={\left[p^{c}\left(1\right),\ldots ,p^{c}\left(m\right)\right]}^{T}$. Recall that per Definition 4, all agents in a given class are identical; thus, the signal models and state priors are the same within the class. Let $U_{c,k}\in \left\{1,2,\ldots ;b\right\}$ be the decision made by the *k*th agent in class *c* distributed according to $p^{c,k}\left(u|y\right)$, with $\gamma_{k}^{c}$ being the rule of the *k*th agent in class *c* (again assuming that, without loss of generality, the decision space {1,2,…;b} is the same for all classes). The message of the *k*th agent in class *c* received by the fusion center is denoted as $X_{c,k}$ and distributed according to $p^{c}\left(x|u,q_{1}^{n},\ldots ;q_{n_{c}}^{n}\right)$. Again, because agents in the same class are identical, the channel model is the same throughout the class. Moreover, let $X_{c}={\left[X_{c,1},\ldots ;X_{c,c_{k}}\right]}^{T}$ be the vector of received messages from all agents in class *c*. We can then extend Assumption 5 to the case of *c* classes.

**Assumption** **6.**
*The following assumptions hold for all classes. Hence, for notational simplicity, when referring to class c we remove the k superscript.*
*(a)* 
*The hypothesis model obeys the following.*

(33)
$$\lim_{n\to \infty }\frac{1}{n}\log\min_{q_{1}^{n},\ldots ,q_{n_{c}}^{n}}\min\left\{p\left(H=1|q_{1}^{n},\ldots ,q_{n_{c}}^{n}\right),p\left(H=0|q_{1}^{n},\ldots ,q_{n_{c}}^{n}\right)\right\}=0.$$

*(b)* 
*The signal model is continuous in $q_{1},\ldots ;q_{n_{c}}$ for all classes, that is, if ${\left\{\alpha_{1,i}\right\}}_{i\in Z},\ldots$; ${\left\{\alpha_{n_{c},i}\right\}}_{i\in Z}$ are sequences in $P^{m}$ such that $\lim_{i\to \infty }\alpha_{j,i}=q_{j}$, $j=1,2,\ldots ;n_{c}$, then ∀y,*

(34)
$$\lim_{i\to \infty }p_{h}^{c}\left(y|\alpha_{1,i},\ldots ;\alpha_{n_{c},i}\right)=p_{h}^{c}\left(y|q_{1},\ldots ;q_{n_{c}}\right),\;h\in \left\{0,1\right\}.$$

*(c)* 
*The channel model is continuous in $q_{1},\ldots ;q_{n_{c}}$ for all classes, that is, if ${\left\{\alpha_{1,i}\right\}}_{i\in Z},\ldots$; ${\left\{\alpha_{n_{c},i}\right\}}_{i\in Z}$ are sequences in $P^{m}$ such that $\lim_{i\to \infty }\alpha_{j,i}=q_{j}$, $j=1,2,\ldots ;n_{c}$, then ∀x, ∀u,*

(35)
$$\lim_{i\to \infty }p^{c}\left(x|u,\alpha_{1,i},\ldots ;\alpha_{n_{c},i}\right)=p^{c}\left(x|u,q_{1},\ldots ;q_{n_{c}}\right)\;h\in \left\{0,1\right\}.$$




The conditions of Assumption 6 closely resemble those of Assumption 5. Namely, Assumption 6.a retains the assumption that the network type for each class should not carry too much information about the hypothesis, while Assumptions 6.b and 6.c extend the assumption that the signal and channel models are continuous in the univariate case to the multi-dimensional case. Before, the models were continuous only in $q_{1}$, whereas we now assume that are continuous in $q_{1},\ldots ;q_{n_{c}}$.

**Lemma** **1.**
*Assume that ∀k and $\lim_{n\to \infty }\frac{c_{k}}{n}>0$; then, under Assumptions 6.a–6.c,*

(36)
$$\Lambda =-\lim_{n\to \infty }\underset{\gamma }{\inf}\min_{\lambda \in [0,1]}\max_{q_{1},\ldots ;q_{n_{c}}}\frac{1}{n}\sum_{c=1}^{n_{c}}-c_{k}D(q_{c}\left|\right|p^{c})+\log\int_{x^{c}}p_{0}{\left(x^{c}|q_{1},\ldots ;q_{n_{c}}\right)}^{1-\lambda }p_{1}{\left(x^{c}|q_{1},\ldots ;q_{n_{c}}\right)}^{\lambda }.$$



Lemma 1 implies that all agents within a given class *c* use the same rule $\gamma^{c}$. When referring to the rule used by all agents in class *c*, we use superscripts to avoid confusion with previously defined notation, where a subscript indicates the rule used by a specific agent. Then, the error exponent takes on a form that allows heterogeneous networks with a high degree of interference to be examined. The details of the extensions of Theorem 1 are provided in Section A.2; Lemma 1 leads to the following theorem.

**Theorem** **2.**
*Let $r_{c}\in \left[0,1\right]$ be the fraction of agents that belong to class $c\in \{1,2,\ldots ;n_{c}\}$, i.e., $c_{k}=\left\lfloor r_{c}n\right\rfloor$, with $\sum_{c=1}^{n_{c}}r_{c}=1$ and where ⌊x⌋ denotes the largest integer that is less than or equal to x. Moreover, suppose that all $r_{c}$ are held constant as n→∞ and that agents in the same class use a common rule. Then, under Assumptions 6.a–6.c,*

(37)
$$\Lambda =-\underset{\gamma^{1},\ldots ;\gamma^{n_{c}}}{\inf}\min_{\lambda \in [0,1]}\max_{q_{1},\ldots ;q_{n_{c}}}\sum_{c=1}^{n_{c}}r_{c}\left[-D(q_{c}\left|\right|p^{c})+\log\int_{x}p_{0}^{c,1}{\left(x|q_{1},\ldots ;q_{n_{c}}\right)}^{1-\lambda }p_{1}^{c,1}{\left(x|q_{1},\ldots ;q_{n_{c}}\right)}^{\lambda }\right].$$



Because identical agents with a common rule may not be optimal, Theorem 2 provides a lower bound on the optimal error exponent. We highlight several important points of Theorem 2 below:Observe that all agents are coupled through the distributions $q_{1},\ldots ;q_{n_{c}}$, and recall that for a given class *c*, $q_{c}$ depends on *all agents in class c* through their states $S_{c,k}$. Hence, the distributions $q_{1},\ldots ;q_{n_{c}}$ collectively depend on all agents in the network, meaning that the received signal, decision, and message for a given agent are dependent on *all agents in the network*. As a result, Theorem 2 captures a very strong form of coupling.Note that the expression in Theorem 2 is not expressed as a limit, does not depend on *n*, and does not depend on the actual size of the classes. Hence, Theorem 2 provides an objective function that can be used to design rules $\gamma^{1},\ldots ;\gamma^{n_{c}}$ *that do not depend on the size of the network*.Theorem 2 depends only on the ratios of the classes; that is, Theorem 2 provides an explicit objective function to find the optimal ratios for asymptotically large networks. Specifically, to find the optimal ratios we can solve
(38)$$\min_{r_{1},\ldots ;r_{n_{c}}}\underset{\gamma^{1},\ldots ;\gamma^{n_{c}}}{\inf}\min_{\lambda \in [0,1]}\max_{q_{1},\ldots ;q_{n_{c}}}\sum_{c=1}^{n_{c}}r_{c}\left[-D(q_{c}\left|\right|p^{c})+\log\int_{x}p_{0}^{c,1}{\left(x|q_{1},\ldots ;q_{n_{c}}\right)}^{1-\lambda }p_{1}^{c,1}{\left(x|q_{1},\ldots ;q_{n_{c}}\right)}^{\lambda }\right].$$In the next section, we present a numerical example that highlights the utility of the proposed framework.

## 5. Numerical Example

We design an example that highlights the different forms of coupling captured by our framework. Note that the total number of agents is never specified, as it is only the fraction of agents in each class (ratio) that matters. However, considering our asymptotic analysis, the network size must be sufficiently large. Consider a three-class system where all agents take one of two states (1 or 2) with $p^{1}\left(S=1\right)=0.5$ and $p^{2}\left(S=1\right)=p^{3}\left(S=1\right)=0.9$, under each hypothesis all classes observe a Gaussian random variable with signal models
(39)$$\begin{array}{c} p_{h}^{1}\left(y|q_{1},q_{2},q_{3}\right)=\frac{1}{\sqrt{2\pi }}\exp\left(-\frac{1}{2}{\left(y-\mu (h,q_{2})\right)}^{2}\right),\;\left[\begin{array}{c} \mu (0,q_{2})\\ \mu (1,q_{2}) \end{array}\right]=\left[\begin{array}{c} 0\\ \alpha r_{1}q_{2}\left(1\right) \end{array}\right] \end{array}$$
and
(40)$$p_{h}^{c}\left(y|q_{1},q_{2},q_{3}\right)=\frac{1}{\sqrt{2\pi }}\exp\left(-\frac{1}{2}y^{2}\right),\;c\in \left\{2,3\right\},$$
where $\mu (h,q_{2})$ is the mean of the signal model when H=h∈{0,1}, $q_{2}$ is the empirical distribution of Class 2, α is a constant that determines the separation between the means of the two hypotheses, and $r_{i}=i_{k}/\left(\sum_{c=1}^{3}c_{k}\right)$ is the ratio for Class *i*.

Important notes about the signal models are as follows:When H=1, the signal model for Class 1 depends only on the number of agents in Class 2 that are in State 1.The signal models for Classes 2 and 3 are constant with respect to the underlying hypothesis as well as the distributions $q_{1}$, $q_{2}$, and $q_{3}$; hence, agents in Class 2 or 3 cannot distinguish between the two hypotheses.
Upon receiving the signal, each agent in class 1 makes a binary decision according to a threshold test, i.e., $u_{1,k}=1\Leftrightarrow y_{1,k}\leq \tau$. Observe that because agents in the other two classes cannot distinguish between hypotheses, their decisions do not matter. Note that the agents belonging to class 1 use identical thresholds; while these may not be optimal, they simplify both design and analysis. Each agent then sends its decision over a binary symmetric channel with the following crossover probability:(41)$$\begin{array}{cc} p^{c}\left(x=1|u=0,q_{1},q_{2},q_{3}\right) & =p^{c}\left(x=0|u=1,q_{1},q_{2},q_{3}\right)\\ & =\max\left\{|\frac{1}{2}-r_{3}q_{3}\left(1\right)|,\rho \right\}\;c\in \left\{1,2,3\right\},\;0<\rho <\frac{1}{2}. \end{array}$$ The parameter ρ governs the minimum achievable crossover probability of the channel. Note that because $|\frac{1}{2}-r_{3}q_{3}\left(1\right)|\leq \frac{1}{2}$, the crossover probability can never be lower than ρ; thus, as ρ increases the channel becomes worse. It can be seen that while Class 2 aids Class 1 in distinguishing between the two hypotheses, Class 3 controls the quality of the channel between the agents and the fusion center. Moreover, if $r_{2}=0$ then agents cannot distinguish the two hypotheses; thus, the error exponent is zero. Similarly, if $r_{3}=0$, the crossover probability for all channels becomes $\frac{1}{2}$; thus, the channel output becomes random and the error exponent becomes zero. This example underscores the impact of cross-class interference on proper optimization of the system. To determine the optimal class ratios, we can solve
(42)$$\left(r_{1}^{*},r_{2}^{*},r_{3}^{*}\right)=\arg\min_{r_{1},r_{2},r_{2}}\max_{q_{1},q_{2},q_{3}}\sum_{c=1}^{3}r_{c}\left[-D(q_{c}\left|\right|p^{c})+\log\sum_{x=1}^{2}\sqrt{p_{0}^{c}\left(x|q_{1},q_{2}\right)p_{1}^{c}\left(x|q_{1},q_{2}\right)}\right],$$
with $r_{1}^{*}+r_{2}^{*}+r_{3}^{*}=1$. For computational simplicity, we set τ=15 and $\lambda =\frac{1}{2}.$ These values can be further optimized.

In Figure 2a, we compute the optimal error exponent as a function of the channel quality ρ for various values of α. Note that the class ratios are optimized for each data point. Recall that as ρ increases, so does the interference, causing the channel to worsen. The importance of the channel on the overall system performance can be clearly seen. As ρ increases, the minimum achievable crossover probability increases and the best-case quality of the channel decreases; hence, the optimal error exponent decreases along with the quality of the channel. In fact, when ρ=0.4, the optimal error exponent is 0.0136, an entire order of magnitude less than when ρ=0.1. The impact of the signal mean for Class 1 is determined by α. Not surprisingly, as the mean increases, the error exponent increases as well; however, we begin to see diminishing returns as we move from α=100 to α=150.

In Figure 2b, the optimal ratio between the three classes is determined as a function of channel quality when α=150. Figure 2b reveals the impact of cross-class interactions. Recall that each class serves a different purpose; Class 1 is the only class that can distinguish between hypotheses, Class 2 controls the sensing capabilities of Class 1, and Class 3 controls the channel quality for Class 1. Hence, the performance of the system relies on the interactions between the three classes. In particular, as ρ increases Class 3 becomes less important to the overall system, as the quality of the channel degrades. This can be seen in Figure 2b by the decreasing $r_{3}^{*}$ and the fact that Class 1 becomes more important to the system, hence the increasing $r_{1}^{*}$.

Finally, we examine the optimizing distribution for computing the error exponents when α=100. As previously noted, the true class distributions of the states ($p^{c}$) do not necessarily dominate asymptotic performance. This can be seen in Figure 2c, which shows that the optimal types are sometimes different from the true distributions. Recall that under S=1 we have $p^{1}=0.5$ and $p^{2}=p^{3}=039$; thus, in this three-class example, it is only when ρ=0.06 that we see the optimizing distribution aligning with the true distribution. We underscore that the network type converges to the true state distribution. Recall that we assume the signal and channel models to be continuous; hence, as the network types converge to the true distributions, the performances of all other distributions in a neighborhood around the true distributions are relatively close. Then, it may be beneficial to design the rule γ to optimize detection for a distribution *close to the true distributions*, as the performance difference is small. This trade-off is captured by our result, where the closeness to $p^{c}$ is captured by the KL divergence and the asymptotic detection performance is captured by the Chernoff information term. Hence, the dominating distribution is the one that offers the best trade-off.

## 6. Proofs

### 6.1. Proof of Theorem 1

Before we begin the proof, we must introduce a number of important definitions and lemmas. There are two sets of lemmas. The first set of lemmas is a series of well-known mathematical facts. Because these are not our contributions but are necessary for the proof of Theorem 1, we omit the proofs, though we provide appropriate citations as necessary. The second set of lemmas is a series of results that, while necessary, are not major contributions of this work; these proofs are provided in Section A.1.

#### 6.1.1. Definitions

**Definition** **5.***A family of functions F defined on a common domain is ****equicontinuous*** *at a point $x_{o}$ if for any ϵ>0 there exists a δ>0 (possibly a function of ϵ and $x_{o}$) such that whenever $|x-x_{o}|<\delta$ we have $|f\left(x\right)-f\left(x_{o}\right)|<\epsilon$ for all f∈F.*

Observe that while the δ above may depend on ϵ and the specific point $x_{o}$, it is not allowed to depend on the specific function *f*, i.e., the chosen δ must work for all functions in F. The next definition removes the dependence on $x_{o}$.

**Definition** **6.***A family of functions F is* ***uniformly equicontinuous*** *if for any ϵ>0 there exists a δ>0 (possibly a function of ϵ) such that whenever |x−y|<δ we have |f(x)−f(y)|<ϵ for all f∈F.*

The above definition states that the same δ must work for all functions f∈F at all points in the domain.

**Definition** **7.***Given a family of Lebesgue measurable functions F with $\int_{x}\left|f\left(x\right)\right|<\infty$ for all f(x)∈F, the integrals $\int_{x}f\left(x\right)$ are* ***uniformly absolutely continuous*** *if ∀ϵ>0 and ∃δ>0 such that for all Lebesgue measurable sets A with ν(A)<δ*(43)$$\int_{A}\left|f\left(x\right)\right|<\epsilon ,$$*for all f∈F, where ν denotes the Lebesgue measure. Of course, these definitions can be extended to any general measure space; however, we focus on the Lebesque measure here for simplicity and to avoid endlessly defining notation. For a thorough discussion of abstract measure spaces, see [49].*

Again, it is important to distinguish that the same δ must work for all functions f∈F for a given ϵ.

**Definition** **8.***Assume that we have a family of measurable functions F with $\int_{x}\left|f\left(x\right)\right|<\infty$ for all f∈F. Moreover, define $I_{a}=\left[-a,a\right]$. Then, the integrals $\int_{x}f\left(x\right)$ are said to be* ***uniformly absolutely convergent*** *if*(44)$$\lim_{a\to \infty }\int_{I_{a}}\left|f\left(x\right)\right|=\int_{x}\left|f\left(x\right)\right|,$$*uniformly in F.*

This is a powerful property, stating that for a given ϵ>0 there is a large enough *a* that all functions in F satisfy
(45)$$|\int_{I_{a}}\left|f\left(x\right)\right|-\int_{x}\left|f\left(x\right)\right||<\epsilon .$$

#### 6.1.2. Key Lemmas

The following lemmas are needed to prove Theorem 1. However, because most are simply known mathematical facts (except Lemma 3, the proof of which is provided in Section A.1), we omit the proofs.

**Lemma** **2.**
*Let F be an equicontinuous and pointwise-bounded family of functions defined on a common domain D. If D is compact, then F is uniformly equicontinuous on D.*


Observe that $P^{m}$ is compact due to it being closed and bounded; because all of our functions (signal models, channel models, etc.) are defined on this space, Lemma 2 allows us to simplify the proof.

**Lemma** **3.**
*Let F and G be families of equicontinuous strictly positive functions defined on a common domain D; furthermore, assume that for each point x∈D we have $\inf_{f\in F}f\left(x\right)>0$, $\inf_{g\in G}g\left(x\right)>0$, $\sup_{f\in F}f\left(x\right)<\infty$, and $\sup_{g\in G}g\left(x\right)<\infty$. Then, the family ${\left\{f{\left(x\right)}^{\lambda }g{\left(x\right)}^{1-\lambda }\right\}}_{f,g,\lambda }$ for f∈F, g∈G, and λ∈[0,1] is equicontinuous on D.*


The next lemma is taken from [49], Theorem 21.

**Lemma** **4.**
*Let $\left\{f_{i}\right\}$ be a sequence of real measurable functions with $\int_{x}\left|f_{i}\left(x\right)\right|<\infty$. Assume that the integrals $\int_{x}f_{i}\left(x\right)$ are uniformly absolutely continuous and uniformly absolutely convergent. Moreover, assume that $f_{i}\to f$ almost everywhere (a.e.); then, $\int_{x}f\left(x\right)<\infty$ and*

(46)
$$\lim_{i\to \infty }\int_{x}\left|f_{i}\left(x\right)-f\left(x\right)\right|=0.$$



Lemma 4 provides a nice immediate result. In particular, suppose we have a function of two variables f(x,y) with $\int_{x}\left|f\left(x,y\right)\right|<\infty$ for all *y* and with $\int_{x}\left|f\left(x,y\right)\right|$ uniformly absolutely continuous and uniformly absolutely convergent with respect to *y*. In this case, Lemma 4 states that the integral $\int_{x}f\left(x,y\right)$ is *continuous in y*. To see this, observe that if $\left\{y_{i}\right\}$ is a sequence with $y_{i}\to y$, then, per the triangle inequality,
(47)$$\lim_{i\to \infty }|\int_{x}f\left(x,y_{i}\right)-\int_{x}f\left(x,y\right)|\leq \lim_{i\to \infty }\int_{x}\left|f\left(x,y_{i}\right)-f\left(x,y\right)\right|=0.$$

### 6.2. Intermediate Lemmas

We next present several intermediate results. The proofs of all these results can be found in Section A.1. Moreover, recalling that we assume all agents to be identical, we consequently omit the *k* superscript in the following lemmas as well as in the proof.

**Lemma** **5.**
*Subject to Assumptions 5.a–5.d, the following two statements hold:*
*(a)* 
*There exists a non-negative function g(x) such that $\int_{x}g\left(x\right)<\infty$ and ∀x, h∈{0,1}, ∀γ, $\forall q\in P^{m}$, and*

(48)
$$\sum_{u}\int_{y}p\left(x|u,q\right)p\left(u|y\right)p_{h}\left(y|q\right)=p_{h}\left(x|q\right)\leq g\left(x\right).$$

*(b)* 
*We have*

(49)
$$\underset{\gamma }{\inf}\min_{\lambda \in [0,1]}\min_{q\in P^{m}}\int_{x}p_{0}{\left(x|q\right)}^{1-\lambda }p_{1}{\left(x|q\right)}^{\lambda }>0.$$




**Lemma** **6.**
*For all ϵ>0, there exists a δ>0 (which depends only on ϵ and h) such that whenever **α** and **β** are two distributions in $P^{m}$ with ${∥\alpha -\beta ∥}_{2}<\delta$, then $\int_{y}\left|p_{h}\left(y|\alpha \right)-p_{h}\left(y|\beta \right)\right|<\epsilon$ for all h∈{0,1}.*


**Lemma** **7.**
*For a fixed x∈X and h∈{0,1}, the family ${\left\{p_{h}\left(x|q\right)2^{-D(q||p)}\right\}}_{\gamma }$ which is indexed by γ is uniformly equicontinuous on $P^{m}$.*


**Lemma** **8.**
*For a fixed x∈X, the family ${\left\{p_{0}{\left(x|q\right)}^{1-\lambda }p_{1}{\left(x|q\right)}^{\lambda }2^{-D(q||p)}\right\}}_{\gamma ,\lambda }$ which is indexed by γ and λ∈[0,1] is uniformly equicontinuous on $P^{m}$.*


**Lemma** **9.**
*For any ϵ>0, there exists a δ>0 (which depends only on ϵ) such that whenever **α** and **β** are two distributions in $P^{m}$ with ${∥\alpha -\beta ∥}_{2}<\delta$, then*

$$\int_{x}\left|p_{0}{\left(x|\alpha \right)}^{1-\lambda }p_{1}{\left(x|\alpha \right)}^{\lambda }2^{-D(\alpha ||p)}-p_{0}{\left(x|\beta \right)}^{1-\lambda }p_{1}{\left(x|\beta \right)}^{\lambda }2^{-D(\beta ||p)}\right|<\epsilon ,$$

*for all γ and λ∈[0,1].*


An immediate consequence of Lemma 9 follows.

**Lemma** **10.**
*For any ϵ>0, there exists a δ>0 (which depends only on ϵ) such that, whenever **α** and **β** are two distributions in $P^{m}$ with ${∥\alpha -\beta ∥}_{2}<\delta$, we have*

(50)
$$|\frac{\int_{x}p_{0}{\left(x|\alpha \right)}^{1-\lambda }p_{1}{\left(x|\alpha \right)}^{\lambda }2^{-D(\alpha ||p)}}{\int_{x}p_{0}{\left(x|\beta \right)}^{1-\lambda }p_{1}{\left(x|\beta \right)}^{\lambda }2^{-D(\beta ||p)}}-1|<\epsilon ,$$

*for all γ and λ∈[0,1].*


The final lemma provides us with a starting point for the proof.

**Lemma** **11.**

$$\Lambda =-\lim_{n\to \infty }\underset{\gamma }{\inf}\min_{\lambda \in [0,1]}\frac{1}{n}\log\int_{x}\sum_{q^{n}}p_{0}{\left(x|q^{n}\right)}^{1-\lambda }p_{1}{\left(x|q^{n}\right)}^{\lambda }p\left(q^{n}\right).$$



Hence, rather than starting directly with the Chernoff information, we start from the expression in Lemma 11. We are now ready to begin the proof.

**Proof of Theorem 1.** Define
(51)$$q^{*}=\arg\max_{q\in P^{m}}-D\left(q||p\right)+\frac{1}{n}\log\int_{x}p_{0}{\left(x|q\right)}^{1-\lambda }p_{1}{\left(x|q\right)}^{\lambda }.$$
and note that $q^{*}$ depends on *n*, γ, and λ; then, for any 0<ϵ<1, per Lemma 10, ∃δ>0, *which depends only on* ϵ, such that whenever $∥q-q^{*}{∥}_{2}<\delta$,
(52)$$|\frac{\int_{x}p_{0}^{k}{\left(x|q\right)}^{1-\lambda }p_{1}^{k}{\left(x|q\right)}^{\lambda }2^{-D(q||p)}}{\int_{x}p_{0}^{k}{\left(x|q^{*}\right)}^{1-\lambda }p_{1}^{k}{\left(x|q^{*}\right)}^{\lambda }2^{-D(q^{*}||p)}}-1|<1-\sqrt{1-\epsilon },$$
for all γ and λ∈[0,1]. Because the agents are identical, they differ only by the rules they use; hence, the same δ works for all agents. For this δ, define
(53)$$T_{\delta }^{n}=\{q^{n}\in Q^{n}:∥q^{n}-q^{*}{∥}_{2}<\delta \},$$
that is, $T_{\delta }^{n}$ is the set of all types that are less than δ away from $q^{*}$ based on the Euclidean distance. There are two important points to make here regarding $T_{\delta }^{n}$:
Because both $Q^{n}$ and $q^{*}$ depend on *n*, $T_{\delta }^{n}$ does as well; however, because δ depends only on ϵ, any type in $T_{\delta }^{n}$ satisfies Equation (Equation 52) regardless of *n* or $q^{*}$.Observe that for any $q\in P^{m}$ there exists a type $q^{n}$ such that $∥q-q^{n}{∥}_{2}<\frac{1}{n}$. Hence, $\exists n_{o}$ such that for all $n\geq n_{o}$ and for any $q\in P^{m}$, $\exists q^{n}$ such that $∥q-q^{n}{∥}_{2}<\delta$. That is, $T_{\delta }^{n}$ is non-empty for all $n\geq n_{o}$. Because $n_{o}$ depends only on δ and δ depends only on ϵ, $n_{o}$ depends only on ϵ, and the same $n_{o}$ *works for all agents and all* λ∈[0,1]. The following argument holds for any $n\geq n_{o}$. We begin by observing that
(54)$$\begin{array}{cc} \frac{\int_{x}p_{0}{\left(x|q^{n}\right)}^{1-\lambda }p_{1}{\left(x|q^{n}\right)}^{\lambda }}{\int_{x}p_{0}{\left(x|q^{*}\right)}^{1-\lambda }p_{1}{\left(x|q^{*}\right)}^{\lambda }2^{-nD(q^{*}||p)}} & =\frac{\int_{x}{\left[\prod_{k}p_{0}^{k}\left(x_{k}|q^{n}\right)\right]}^{1-\lambda }{\left[\prod_{k}p_{1}^{k}\left(x_{k}|q^{n}\right)\right]}^{\lambda }}{\int_{x}{\left[\prod_{k}p_{0}^{k}\left(x_{k}|q^{*}\right)\right]}^{1-\lambda }{\left[\prod_{k}p_{1}^{k}\left(x_{k}|q^{*}\right)\right]}^{\lambda }2^{-nD(q^{*}||p)}} \end{array}$$
(55)$$\begin{array}{cc} \; & \;\;\;\;\;\;\;\;\;=\frac{\int_{x}\prod_{k}p_{0}^{k}{\left(x_{k}|q^{n}\right)}^{1-\lambda }p_{1}^{k}{\left(x_{k}|q^{n}\right)}^{\lambda }}{\int_{x}\prod_{k}p_{0}^{k}{\left(x_{k}|q^{*}\right)}^{1-\lambda }p_{1}^{k}{\left(x_{k}|q^{*}\right)}^{\lambda }2^{-D(q^{*}||p)}} \end{array}$$
(56)$$\begin{array}{cc} \; & \;\;\;\;\;\;\;\;\;=\prod_{k}\frac{\int_{x}p_{0}^{k}{\left(x_{k}|q^{n}\right)}^{1-\lambda }p_{1}^{k}{\left(x_{k}|q^{n}\right)}^{\lambda }}{\int_{x}p_{0}^{k}{\left(x_{k}|q^{*}\right)}^{1-\lambda }p_{1}^{k}{\left(x_{k}|q^{*}\right)}^{\lambda }2^{-D(q^{*}||p)}}. \end{array}$$ Then, we have the following:
(57)$$\begin{array}{cc} \sum_{q^{n}}\frac{\int_{x}p_{0}{\left(x|q^{n}\right)}^{1-\lambda }p_{1}{\left(x|q^{n}\right)}^{\lambda }}{\int_{x}p_{0}{\left(x|q^{*}\right)}^{1-\lambda }p_{1}{\left(x|q^{*}\right)}^{\lambda }2^{-nD(q^{*}||p)}}p\left(q^{n}\right) & =\sum_{q^{n}}\prod_{k}\frac{\int_{x}p_{0}^{k}{\left(x_{k}|q^{n}\right)}^{1-\lambda }p_{1}^{k}{\left(x_{k}|q^{n}\right)}^{\lambda }}{\int_{x}p_{0}^{k}{\left(x_{k}|q^{*}\right)}^{1-\lambda }p_{1}^{k}{\left(x_{k}|q^{*}\right)}^{\lambda }2^{-D(q^{*}||p)}}p\left(q^{n}\right) \end{array}$$
(58)$$\begin{array}{cc} \; & \;\;\;\;\;\;\;\;\;\;\;\;\;\;\;\;\;\;\;\overset{\left(a\right)}{\leq }\sum_{q^{n}}\prod_{k}\frac{\int_{x}p_{0}^{k}{\left(x_{k}|q^{n}\right)}^{1-\lambda }p_{1}^{k}{\left(x_{k}|q^{n}\right)}^{\lambda }}{\int_{x}p_{0}^{k}{\left(x_{k}|q^{*}\right)}^{1-\lambda }p_{1}^{k}{\left(x_{k}|q^{*}\right)}^{\lambda }2^{-D(q^{*}||p)}}2^{-nD(q^{n}||p)} \end{array}$$
(59)$$\begin{array}{cc} \; & \;\;\;\;\;\;\;\;\;\;\;\;\;\;=\sum_{q^{n}}\prod_{k}\frac{\int_{x}p_{0}^{k}{\left(x_{k}|q^{n}\right)}^{1-\lambda }p_{1}^{k}{\left(x_{k}|q^{n}\right)}^{\lambda }2^{-D(q^{n}||p)}}{\int_{x}p_{0}^{k}{\left(x_{k}|q^{*}\right)}^{1-\lambda }p_{1}^{k}{\left(x_{k}|q^{*}\right)}^{\lambda }2^{-D(q^{*}||p)}} \end{array}$$
(60)$$\begin{array}{cc} \; & \;\;\;\;\overset{\left(b\right)}{\leq }\sum_{q^{n}}1\overset{\left(c\right)}{\leq }{\left(n+1\right)}^{m}, \end{array}$$
where (a) holds, as $p\left(q^{n}\right)\leq 2^{-nD(q^{n}||p)}$ [17,50], where (b) is due to the definition of $q^{*}$ and (c) holds because for any *n* the number of types is upper-bounded by ${\left(n+1\right)}^{m}$ ([50], Theorem 11.1.1). Then, taking the *n*-th root yields the upper bound
(61)$${\left[\frac{\sum_{q^{n}}\int_{x}p_{0}{\left(x|q^{n}\right)}^{1-\lambda }p_{1}{\left(x|q^{n}\right)}^{\lambda }p\left(q^{n}\right)}{\int_{x}p_{0}{\left(x|q^{*}\right)}^{1-\lambda }p_{1}{\left(x|q^{*}\right)}^{\lambda }2^{-nD(q^{*}||p)}}\right]}^{\frac{1}{n}}\leq {\left(n+1\right)}^{\frac{m}{n}}.$$ Observe that ${\left(n+1\right)}^{\frac{m}{n}}\to 1$; thus, $\exists n_{1}$ such that $\forall n\geq n_{1}$, ${\left(n+1\right)}^{\frac{m}{n}}\leq 1+\epsilon$. Turning our attention to the lower bound,
(62)$$\begin{array}{c} \sum_{q^{n}}\prod_{k}\frac{\int_{x}p_{0}^{k}{\left(x_{k}|q^{n}\right)}^{1-\lambda }p_{1}^{k}{\left(x_{k}|q^{n}\right)}^{\lambda }}{\int_{x}p_{0}^{k}{\left(x_{k}|q^{*}\right)}^{1-\lambda }p_{1}^{k}{\left(x_{k}|q^{*}\right)}^{\lambda }2^{-D(q^{*}||p)}}p\left(q^{n}\right) \end{array}$$
(63)$$\begin{array}{c} \geq \sum_{q^{n}\in T_{o}^{n}}\prod_{k}\frac{\int_{x}p_{0}^{k}{\left(x_{k}|q^{n}\right)}^{1-\lambda }p_{1}^{k}{\left(x_{k}|q^{n}\right)}^{\lambda }}{\int_{x}p_{0}^{k}{\left(x_{k}|q^{*}\right)}^{1-\lambda }p_{1}^{k}{\left(x_{k}|q^{*}\right)}^{\lambda }2^{-D(q^{*}||p)}}p\left(q^{n}\right) \end{array}$$
(64)$$\begin{array}{cc} \; & \overset{\left(a\right)}{\geq }\frac{1}{{\left(n+1\right)}^{m}}\sum_{q^{n}\in T_{o}^{n}}\prod_{k}\frac{\int_{x}p_{0}^{k}{\left(x_{k}|q^{n}\right)}^{1-\lambda }p_{1}^{k}{\left(x_{k}|q^{n}\right)}^{\lambda }}{\int_{x}p_{0}^{k}{\left(x_{k}|q^{*}\right)}^{1-\lambda }p_{1}^{k}{\left(x_{k}|q^{*}\right)}^{\lambda }2^{-D(q^{*}||p)}}2^{-nD(q^{n}||p)} \end{array}$$
(65)$$\begin{array}{cc} \; & =\frac{1}{{\left(n+1\right)}^{m}}\sum_{q^{n}\in T_{o}^{n}}\prod_{k}\frac{\int_{x}p_{0}^{k}{\left(x_{k}|q^{n}\right)}^{1-\lambda }p_{1}^{k}{\left(x_{k}|q^{n}\right)}^{\lambda }2^{-D(q^{n}||p)}}{\int_{x}p_{0}^{k}{\left(x_{k}|q^{*}\right)}^{1-\lambda }p_{1}^{k}{\left(x_{k}|q^{*}\right)}^{\lambda }2^{-D(q^{*}||p)}} \end{array}$$
(66)$$\begin{array}{cc} \; & \overset{\left(b\right)}{\geq }\frac{1}{{\left(n+1\right)}^{m}}\sum_{q^{n}\in T_{o}^{n}}{\left(1-\epsilon \right)}^{n}\overset{\left(c\right)}{\geq }\frac{1}{{\left(n+1\right)}^{m}}{\left(1-1+\sqrt{1-\epsilon }\right)}^{n}, \end{array}$$
where (a) holds because $p\left(q^{n}\right)\geq \frac{1}{{\left(n+1\right)}^{\left|S\right|}}2^{-nD(q^{n}||p)}$ [17,50], (b) is due to the definition of $T_{o}^{n}$, and (c) holds because $T_{o}^{n}$ is non-empty for $n\geq n_{o}$. Taking the *n*-th root provides
(67)$${\left[\frac{\sum_{q^{n}}\int_{x}p_{0}{\left(x|q^{n}\right)}^{1-\lambda }p_{1}{\left(x|q^{n}\right)}^{\lambda }p\left(q^{n}\right)}{\int_{x}p_{0}{\left(x|q^{*}\right)}^{1-\lambda }p_{1}{\left(x|q^{*}\right)}^{\lambda }2^{-nD(q^{*}||p)}}\right]}^{\frac{1}{n}}\geq {\left(n+1\right)}^{\frac{-m}{n}}\left(1-1+\sqrt{1-\epsilon }\right).$$ Observe that ${\left(n+1\right)}^{\frac{-m}{n}}\to 1$; thus, $\exists n_{2}$ such that $\forall n\geq n_{2}$, ${\left(n+1\right)}^{\frac{-m}{n}}\left(1-1+\sqrt{1-\epsilon }\right)\geq \left(1-1+\sqrt{1-\epsilon }\right)\left(1-1+\sqrt{1-\epsilon }\right)=1-\epsilon$. Then, we can take $n_{\epsilon }=\max\left\{n_{o},n_{1},n_{2}\right\}$, meaning that for all $n\geq n_{\epsilon }$ we have
(68)$$1-\epsilon \leq {\left[\frac{\sum_{q^{n}}\int_{x}p_{0}{\left(x|q^{n}\right)}^{1-\lambda }p_{1}{\left(x|q^{n}\right)}^{\lambda }p\left(q^{n}\right)}{\int_{x}p_{0}{\left(x|q^{*}\right)}^{1-\lambda }p_{1}{\left(x|q^{*}\right)}^{\lambda }2^{-nD(q^{*}||p)}}\right]}^{\frac{1}{n}}\leq 1+\epsilon .$$ Because none of $n_{o}$, $n_{1}$, or $n_{2}$ depend on $q^{*}$, γ, or λ, it is the case that $n_{\epsilon }$ does not depend on $q^{*}$, γ, or λ; hence, we have uniform convergence, which completes the proof. □

### 6.3. Proof of Theorem 2

Because we assume that agents of the same class use the same rule, if we focus on class *c* we have
(69)$$p_{h}\left(x^{c}|q_{1},\ldots ;q_{n_{c}}\right)=\prod_{k=1}^{c_{k}}p^{c,k}\left(x_{c,k}|q_{1}^{n},\ldots ;q_{n_{c}}^{n}\right),$$
which is a consequence of Equations (Equation 5) and (Equation 8). Then, we have
(70)$$\begin{array}{cc} \; & -c_{k}D(q_{c}\left|\right|p^{c})+\log\int_{x}p_{0}{\left(x^{c}|q_{1},\ldots ;q_{n_{c}}\right)}^{1-\lambda }p_{1}{\left(x^{c}|q_{1},\ldots ;q_{n_{c}}\right)}^{\lambda }\\ = & \sum_{k=1}^{c_{k}}-D(q_{c}\left|\right|p^{c})+\log\int_{x}p_{0}^{c,k}{\left(x|q_{1},\ldots ;q_{n_{c}}\right)}^{1-\lambda }p_{1}^{c,k}{\left(x|q_{1},\ldots ;q_{n_{c}}\right)}^{\lambda }, \end{array}$$
with
(71)$$p_{h}^{c,k}\left(x|q_{1},\ldots ;q_{n_{c}}\right)=\sum_{u=1}^{b}p^{c}\left(x|u,q_{1},\ldots ;q_{n_{c}}\right)\int_{y}p^{c,k}\left(u|y\right)p_{h}^{c}\left(y|q_{1},\ldots ;q_{n_{c}}\right),\;h\in \left\{0,1\right\}.$$ If all agents in Class *c* use rule $\gamma^{c}$, then every term in the sum of Equation (Equation 70) is equal. Hence,
(72)$$\begin{array}{cc} \; & \frac{1}{n}\sum_{c=1}^{n_{c}}-c_{k}D(q_{c}\left|\right|p^{c})+\log\int_{x}p_{0}{\left(x^{c}|q_{1},\ldots ;q_{n_{c}}\right)}^{1-\lambda }p_{1}{\left(x^{c}|q_{1},\ldots ;q_{n_{c}}\right)}^{\lambda }\\ = & \sum_{c=1}^{n_{c}}\frac{c_{k}}{n}\left[-D(q_{c}\left|\right|p^{c})+\log\int_{x}p_{0}^{c,1}{\left(x|q_{1},\ldots ;q_{n_{c}}\right)}^{1-\lambda }p_{1}^{c,1}{\left(x|q_{1},\ldots ;q_{n_{c}}\right)}^{\lambda }\right]. \end{array}$$ We now turn our attention to the difference
(73)$$\begin{array}{cc} \sum_{c=1}^{n_{c}}r_{c}[-D(q_{c}\left|\right|p^{c}) & +\log\int_{x}p_{0}^{c,1}{\left(x|q_{1},\ldots ;q_{n_{c}}\right)}^{1-\lambda }p_{1}^{c,1}{\left(x|q_{1},\ldots ;q_{n_{c}}\right)}^{\lambda }]\\ \; & -\sum_{c=1}^{n_{c}}\frac{c_{k}}{n}\left[-D(q_{c}\left|\right|p^{c})+\log\int_{x}p_{0}^{c,1}{\left(x|q_{1},\ldots ;q_{n_{c}}\right)}^{1-\lambda }p_{1}^{c,1}{\left(x|q_{1},\ldots ;q_{n_{c}}\right)}^{\lambda }\right], \end{array}$$
which is equivalent to
(74)$$\sum_{c=1}^{n_{c}}\frac{r_{c}n-\left\lfloor r_{c}n\right\rfloor }{n}\left[-D(q_{c}\left|\right|p^{c})+\log\int_{x}p_{0}^{c,1}{\left(x|q_{1},\ldots ;q_{n_{c}}\right)}^{1-\lambda }p_{1}^{c,1}{\left(x|q_{1},\ldots ;q_{n_{c}}\right)}^{\lambda }\right].$$ Observe that
(75)$$-D(q_{c}\left|\right|p^{c})+\log\int_{x}p_{0}^{c,1}{\left(x|q_{1},\ldots ;q_{n_{c}}\right)}^{1-\lambda }p_{1}^{c,1}{\left(x|q_{1},\ldots ;q_{n_{c}}\right)}^{\lambda }]\leq 0,$$
for all classes, which is a consequence of the non-negativity of the KL divergence [50] and the non-positivity of the Chernoff information [44]. Combining this with the fact that $\frac{r_{c}n-\left\lfloor r_{c}n\right\rfloor }{n}\geq 0$, we see that (Equation 74) is upper-bounded by zero. For a lower bound, observe that $\frac{r_{c}n-\left\lfloor r_{c}n\right\rfloor }{n}\leq \frac{1}{n}$, which yields the result that Equation (Equation 74) is lower-bounded by
(76)$$\begin{array}{c} \sum_{c=1}^{n_{c}}\frac{1}{n}\left[-D(q_{c}\left|\right|p^{c})+\log\int_{x}p_{0}^{c,1}{\left(x|q_{1},\ldots ;q_{n_{c}}\right)}^{1-\lambda }p_{1}^{c,1}{\left(x|q_{1},\ldots ;q_{n_{c}}\right)}^{\lambda }\right] \end{array}$$
(77)$$\begin{array}{c} \geq \sum_{c=1}^{n_{c}}\frac{1}{n}\left[-\max_{q}D(q||p^{c})+\underset{\gamma }{\inf}\min_{\lambda \in [0,1]}\min_{q_{1},\ldots ;q_{n_{c}}}\log\int_{x}p_{0}^{c,1}{\left(x|q_{1},\ldots ;q_{n_{c}}\right)}^{1-\lambda }p_{1}^{c,1}{\left(x|q_{1},\ldots ;q_{n_{c}}\right)}^{\lambda }\right]. \end{array}$$ The KL divergence (for finite alphabets) is bounded, and repeating the proof of Lemma 5 for the multi-class case using Assumptions 6.b and 6.c guarantees that the logarithm terms are finite. Hence, Equation (77) goes to zero as n→∞. Moreover, note that this lower bound is independent of the strategies γ and λ and the distributions $q_{1},\ldots ;q_{n_{c}}$. This means that Equation (Equation 74) converges uniformly in γ and λ and the distributions $q_{1},\ldots ;q_{n_{c}}$, which allows us to take the infimum, minimum, and maximum, respectively. To see this, observe that the upper bound provides
(78)$$\begin{array}{cc} \max_{q_{1},\ldots ;q_{n_{c}}} & \sum_{c=1}^{n_{c}}\frac{c_{k}}{n}\left[-D(q_{c}\left|\right|p^{c})+\log\int_{x}p_{0}^{c,1}{\left(x|q_{1},\ldots ;q_{n_{c}}\right)}^{1-\lambda }p_{1}^{c,1}{\left(x|q_{1},\ldots ;q_{n_{c}}\right)}^{\lambda }\right]\\ \geq & \sum_{c=1}^{n_{c}}\frac{c_{k}}{n}\left[-D(q_{c}\left|\right|p^{c})+\log\int_{x}p_{0}^{c,1}{\left(x|q_{1},\ldots ;q_{n_{c}}\right)}^{1-\lambda }p_{1}^{c,1}{\left(x|q_{1},\ldots ;q_{n_{c}}\right)}^{\lambda }\right]\\ \geq & \sum_{c=1}^{n_{c}}r_{c}\left[-D(q_{c}\left|\right|p^{c})+\log\int_{x}p_{0}^{c,1}{\left(x|q_{1},\ldots ;q_{n_{c}}\right)}^{1-\lambda }p_{1}^{c,1}{\left(x|q_{1},\ldots ;q_{n_{c}}\right)}^{\lambda }\right]. \end{array}$$ As this is true for all $q_{1},\ldots ;q_{n_{c}}$, we have
(79)$$\begin{array}{cc} \; & \max_{q_{1},\ldots ;q_{n_{c}}}\sum_{c=1}^{n_{c}}\frac{c_{k}}{n}\left[-D(q_{c}\left|\right|p^{c})+\log\int_{x}p_{0}^{c,1}{\left(x|q_{1},\ldots ;q_{n_{c}}\right)}^{1-\lambda }p_{1}^{c,1}{\left(x|q_{1},\ldots ;q_{n_{c}}\right)}^{\lambda }\right]\\ \geq & \max_{q_{1},\ldots ;q_{n_{c}}}\sum_{c=1}^{n_{c}}r_{c}\left[-D(q_{c}\left|\right|p^{c})+\log\int_{x}p_{0}^{c,1}{\left(x|q_{1},\ldots ;q_{n_{c}}\right)}^{1-\lambda }p_{1}^{c,1}{\left(x|q_{1},\ldots ;q_{n_{c}}\right)}^{\lambda }\right]. \end{array}$$ The same argument can be repeated to obtain
(80)$$\begin{array}{cc} \; & \max_{q_{1},\ldots ;q_{n_{c}}}\sum_{c=1}^{n_{c}}r_{c}\left[-D(q_{c}\left|\right|p^{c})+\log\int_{x}p_{0}^{c,1}{\left(x|q_{1},\ldots ;q_{n_{c}}\right)}^{1-\lambda }p_{1}^{c,1}{\left(x|q_{1},\ldots ;q_{n_{c}}\right)}^{\lambda }\right]\\ \geq & \max_{q_{1},\ldots ;q_{n_{c}}}\sum_{c=1}^{n_{c}}\frac{c_{k}}{n}\left[-D(q_{c}\left|\right|p^{c})+\log\int_{x}p_{0}^{c,1}{\left(x|q_{1},\ldots ;q_{n_{c}}\right)}^{1-\lambda }p_{1}^{c,1}{\left(x|q_{1},\ldots ;q_{n_{c}}\right)}^{\lambda }\right]\\ + & \sum_{c=1}^{n_{c}}\frac{1}{n}\left[-\max_{q}D(q||p^{c})+\underset{\gamma }{\inf}\min_{\lambda \in [0,1]}\min_{q_{1},\ldots ;q_{n_{c}}}\log\int_{x}p_{0}^{c,1}{\left(x|q_{1},\ldots ;q_{n_{c}}\right)}^{1-\lambda }p_{1}^{c,1}{\left(x|q_{1},\ldots ;q_{n_{c}}\right)}^{\lambda }\right]. \end{array}$$ Hence, the difference
(81)$$\begin{array}{cc} \max_{q_{1},\ldots ;q_{n_{c}}}\sum_{c=1}^{n_{c}}r_{c}[- & D(q_{c}\left|\right|p^{c})+\log\int_{x}p_{0}^{c,1}{\left(x|q_{1},\ldots ;q_{n_{c}}\right)}^{1-\lambda }p_{1}^{c,1}{\left(x|q_{1},\ldots ;q_{n_{c}}\right)}^{\lambda }]\\ \; & -\max_{q_{1},\ldots ;q_{n_{c}}}\sum_{c=1}^{n_{c}}\frac{c_{k}}{n}\left[-D(q_{c}\left|\right|p^{c})+\log\int_{x}p_{0}^{c,1}{\left(x|q_{1},\ldots ;q_{n_{c}}\right)}^{1-\lambda }p_{1}^{c,1}{\left(x|q_{1},\ldots ;q_{n_{c}}\right)}^{\lambda }\right], \end{array}$$
goes to zero as n→∞. Repeating the same argument with the minimum over λ followed by the infimum over γ completes the proof.

## 7. Conclusions

In this paper, we have introduced a new framework for decentralized inference that captures a high degree of coupling between the agents. Under our framework, the empirical distribution of the network state induces a global coupling across agents. We find an asymptotically equivalent expression to the Chernoff information and unveil a number of interesting properties, such as the fact that the true state distribution does not always dominate asymptotic performance. For the multi-class case, we characterize how ratios of classes of agents affect performance. We further allow for a lossy communication link between the agents and the fusion center and investigate the effects of the channel on overall performance. Our work extends prior work on distributed detection, and is able to break the requirement of conditionally independent observations when correlation is present. In future work, we will remove the fusion center from the system and require agents to directly communicate with each other, as in a purely decentralized ad hoc system. In addition, we will consider the introduction of actions by the agents which can affect observations by other agents to enable the consideration of active hypothesis testing in a distributed setting.

## Figures and Tables

**Figure 1 entropy-25-01313-f001:**
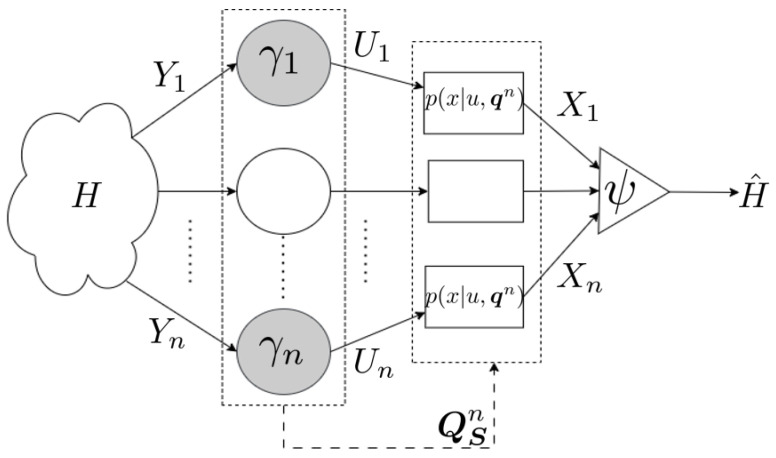
A set of *n* agents receive signals $Y_{k}$ and states $S_{k}$. Each agent is characterized by a decision rule $\gamma_{k}$ and sends a message $X_{k}$ to the fusion center, which outputs $\hat{H}$. The empirical distribution of the states $Q_{S}^{n}$ governs the behavior of the signals $Y_{k}$ as well as the communication channels.

**Figure 2 entropy-25-01313-f002:**
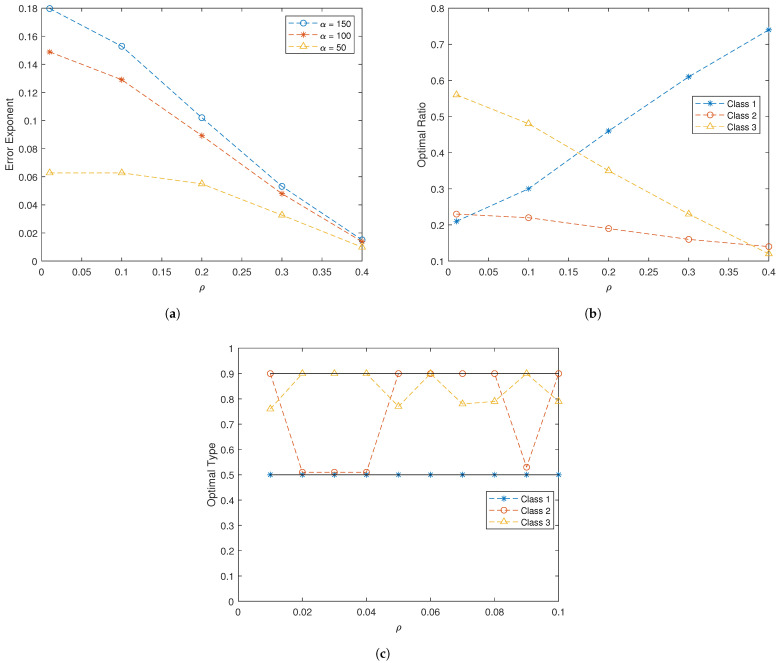
Three-class example with coupled signaling and state-dependent channels: (**a**) the optimal error exponent as a function of ρ, highlighting the importance of the channel on the overall system; (**b**) the optimal class ratios for α=150 (as ρ increases, Class 3 becomes less important to the overall system); and (**c**) the dominating distributions α=100, which may be different from the true distributions.

## Data Availability

No new data were created or analyzed in this study. Data sharing is not applicable to this article.

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
