# Peer review of "Joint Detection and Communication over Type-Sensitive Networks"

_entropy, 2023, doi:10.3390/e25091313_

Round 1
Reviewer 1 Report
Specific review comments
1) What is the overlap of this paper with [16] that is a submitted paper of same authors on same subject? What is the significance of this paper over [16]? Need to emphasize this issue better.
2) It would be desirable to provide a discussion with some insights to when the limit in (10) exits, and when it does not.
3) It would be desirable to provide a discussion concerning the conditions in Definition 2.
4) On the bottom of p. 7, the first paragraph of section 4 provides a discussion on Theorem 1, which is presented only on lower half of page 8. So the reader does not even know what this theorem says at this point. Please first present the theorem and only after that provide a discussion of this theorem.
5) I could not understand the material on p. 9 included in point 3. A better presentation is desirable.
6) There is also a writing style problem in this paper that makes it too long. For example on p. 10 just before (36) the authors include the statement “We shall next focus on the quantity” followed by (36). All this can be removed since it does not add any information in view of the next statement “To prove the result, we wish to show that” followed by (37) and (38).
7) On p. 10 after (42) the authors state “Then, the main challenge of the proof is to show Equation (42)”. Please indicate where this is shown in the paper.
8) All the conditions in Assumption 6 on p. 11 have to discussed emphasizing their implications.
9) Please indicate where Lemma 1 from p. 12 is proven.
10) Numerical results consider only a two classes case. Taking into account that section 4.2 considers the multi-class case, it would be desirable to present also a numerical example for a multi-class case.
11) Fig. 4 (d) illustrates that even when the channel quality improves, the effect of knowing the on the error exponent is small. Need to explain in more detail why this is so. Is channel knowledge important or not ? This issue should be checked for a larger set of channels and systems such that we can learn something.
In general the English is acceptable. However a better organization of the paper is necessary/ Please see specific review comments
Reviewer 2 Report
The authors have presented very interesting research. Authors are advised to answer few questions.
1) Please explain the need of the current approach.
2) The novelty is not clear.
3) The abstract can be rewritten with more technical aspects.
4) The conclusion should scope the future work more clearly.
Minor English improvement
Reviewer 3 Report
This paper investigates the problem of distributed decision making in a simple network with finite states. The quality of presentation is moderate. The authors have given a rather concrete formulation (with some annoying mistakes). The analysis is given for homogeneous network. The problem gets quite challenging to track in heterogeneous networks. The author hence attack the problem only by describing the extension of their analysis of homogeneous setting.
All of all the paper seems to be technically qualified for publication; however, it needs to be majorly revised to be publishable. There are also some major concerns that needed to be addressed:
A. As I can see in the references, there are several related publications of the authors, some of then only been submitted. The authors need to clarify, what is their contribution as compared to them. In particular, they need to indicate how is this work different from their recent submission to Trans-IT (ref. 16). This is quite crucial.
B. The literature review should be updated. The author have mostly cited their recent publications and the other references are rather old. Maybe, the authors could also cite some recent publications on distributed inference.
C. The paper has quite lots of annoying mistakes. Several notations are used inconsistent. Some particular parameters (like priors or likelihoods) have not been defined. Equations are not properly aligned and at some point later in the paper the flow of the discussions is lost.
All of all, I suggest that the authors revise their paper majorly. Please find some other minor comments below.
Some minor comments:
1. To me Assumption 1 sounds quite intuitive. Maybe, a bit of discussion in this regard could be given in the paper.
2. Why the channel model also depends on the type? I can think of some examples of sequence transmission with asymmetric DMC; however, for the one-shot transmission as here, I don't see it immediately. Maybe, you could give some short illustration.
3. At some point after Assumption 4, the authors state that "Upon receiving X and Q_S^n$. This is not clear to me how "Q_S^n" is received. Maybe I missed it somewhere, but I took it as assumption afterwards. The authors however need to clarify it in the revised version. In that case, the former comment on the channel transition is also addressed, since by sending Q_S^n, the channel shall be dependent on it as the input.
4. I would rather prefer adding a randomizer also in the argument of decision rule. But it is only the matter of taste.
5. Unless I am missing it somewhere, \pi_i(q) does not seem to be defined (used at Eq. (9)). I believe it is p(q|H=i), but it needs to be defined!
6. The discussions on Assumption 5 are very useful. Only one minor comment: for (f), as they stated, it seems to be always correct, but couldn't the author prove it for a general case? A quick look tells me that it could be somehow related to exponential function of the KL-divergence. Can't one use the Gibbs' inequality then to show it? Not sure though!
7. Please fix equations (20)-(23) they are totally out of margin.
8. Please fix Figure 2.
9. Equations (73)-(75) have same issue! Also in Appendix
Please refer to my main comments. The writing is fine by itself. It's only the problem of presentation that should be addressed in the revision.
Round 2
Reviewer 3 Report
The authors have given detailed response to my comments that is very appreciated. They have further put good efforts to revise the paper. The revised version reads very much improved as compared to the initial version. Also my concerns are pretty much addressed.
I believe the current version, after a minor revision for few remaining writing errors, is ready for publication.